# Interband resonant high-harmonic generation by valley polarized electron–hole pairs

Naotaka Yoshikawa [1,2], Kohei Nagai[1], Kento Uchida[1], Yuhei Takaguchi[3], Shogo Sasaki[3], Yasumitsu Miyata[3] & Koichiro Tanaka [1,2]

High-harmonic generation in solids is a unique tool to investigate the electron dynamics in strong light fields. The systematic study in monolayer materials is required to deepen the insight into the fundamental mechanism of high-harmonic generation. Here we demonstrated nonperturbative high harmonics up to 18th order in monolayer transition metal dichalcogenides. We found the enhancement in the even-order high harmonics which is attributed to the resonance to the band nesting energy. The symmetry analysis shows that the valley polarization and anisotropic band structure lead to polarization of the high-harmonic radiation. The calculation based on the three-step model in solids revealed that the electron–hole polarization driven to the band nesting region should contribute to the high harmonic radiation, where the electrons and holes generated at neighboring lattice sites are taken into account. Our findings open the way for attosecond science with monolayer materials having widely tunable electronic structures.

[1] Department of Physics, Graduate School of Science, Kyoto University, Sakyo-ku, Kyoto 606-8502, Japan. [2] Institute for Integrated Cell-Material Sciences (WPI-iCeMS), Kyoto University, Sakyo-ku, Kyoto 606-8501, Japan. [3] Department of Physics, Tokyo Metropolitan University, Hachioji, Tokyo 192-0397, Japan. Correspondence and requests for materials should be addressed to N.Y. (email: yoshikawa@thz.phys.s.u-tokyo.ac.jp) or to K.T. (email: kochan@scphys.kyoto-u.ac.jp)

Research on generation of high harmonics (HHG) from atomic or molecular gases has produced coherent broadband radiation in the extreme ultraviolet region and attosecond pulses[1,2]. Moreover, HHG has recently been reported from several solid-state materials[3–12]. Nonperturbative HHG in solids can be achieved with a driving laser with a peak intensity as low as $10^{12}$ W cm$^{-2}$, which is much smaller than that required in the gas phase, suggesting the potential for a compact, highly efficient attosecond light source. In addition, solid-state HHG can capture the electronic properties of materials as tomographic imaging of molecular gases has been demonstrated by high-harmonic spectroscopy[13–15]. High harmonic spectroscopy in solids shows the possibility of probing the electronic band structure as it relates to crystal symmetry and interatomic bonding[3,4,7,10,11]. A unified understanding of the mechanism of HHG in solids is necessary for the application of high harmonic spectroscopy. Although several theoretical models have been proposed for solid-state HHG in terms of interband polarization and intraband electron dynamics as well as their interplay[3–11,16–19], the underlying mechanism is still under debate. It will be important to gain an understanding of HHG in single-atomic-layer solids, because propagation effects such as the phase matching condition, which would otherwise obscure intrinsic features of HHG in bulk crystals, do not affect ideal two-dimensional systems.

HHG from monolayer graphene was recently reported[20–22], and its observed anomalous ellipticity dependence was able to be reproduced by a fully quantum mechanical theory in which both intraband and interband contributions are included[18]. HHG was also observed from monolayer MoS$_2$, which is one of the monolayer transition metal dichalcogenides (TMDs)[23]. Monolayer TMDs, which have a finite bandgap in contrast to graphene, have attracted much attention for their exotic properties such as enhancement of luminescence derived from their indirect-to-direct bandgap transition from bulk to monolayer form[24,25], extremely large exciton binding energy[26], and valley pseudospin physics arising from inversion symmetry breaking and a large spin-orbit interaction[27,28]. Liu et al. found that the even-order high harmonics are predominantly polarized perpendicular to the pump laser and attributed this to the intraband anomalous transverse current arising from the material's Berry curvature[23]. On the other hand, a recent experiment with an optical pump showed that the interband contribution dominates the intraband one in bulk ZnO[29]. Liu et al. also showed that the even orders of HHG are enhanced at much higher energies compared with the lowest exciton or bandgap energy, while the odd orders are monotonically suppressed at higher energies. The enhancement at higher orders and its relation to the perpendicular polarization of even-order HHG is not yet understood.

In this study, we systematically investigated HHG from four kinds of monolayer TMDs with different bandgap energies (MoSe$_2$, WSe$_2$, MoS$_2$, and WS$_2$) to examine the resonance effect and polarization of HHG in monolayer TMDs. By comparing the HHG and optical absorption spectra of the monolayer TMDs, we found that HHG is enhanced when it is in resonance with the optical transition due to band nesting, indicating that the interband polarization mainly contributes to the even-order HHG. A simple calculation that is an extension of the three-step model for electron–hole pairs in solids shows that the electron–hole pairs driven to the band nesting region significantly contributes to the resonance enhancement, and anisotropic driving of electron–hole polarizations around the $K$ and $K'$ points determines polarization selection rule of HHG.

## Results

**High harmonic spectra from monolayer TMDs**. We excited the monolayer TMDs by using mid-infrared pulses (0.26 eV photon energy, 1.7 TWcm$^{-2}$ peak intensity) and detected HHG in the near-infrared to ultraviolet energy region (see the Methods and Supplementary Figure 1). Figure 1 shows the high harmonic spectra from fifth to sixteenth order of (a) MoSe$_2$, (b) WSe$_2$, (c) MoS$_2$, and (d) WS$_2$ monolayers (see the Methods and Supplementary Figure 2 for the full spectrum up to 18th harmonics). We set the polarization of the mid-infrared pulses in parallel with the zigzag direction (M–M (M=Mo, W) direction) (see Supplementary Note 1). The polarization of the even-order harmonic radiation was perpendicular to that of the incident laser light, while the polarization of the odd-order harmonics was parallel (Supplementary Figure 3a)[23]. The excitation power dependence shows the nonperturbative nature of HHG from monolayer TMDs (see Supplementary Note 2).

**Interband resonant enhancement of HHG**. Figure 2a–d shows the polarization-unresolved emission intensities of the even-order harmonics, and Fig. 2e–h the optical absorption spectra. In the absorption spectra, the peaks labeled A and B are attributed to so-called A and B excitons, consisting of electrons and holes localized in the $K$ valleys in momentum space. In addition, the band structures of the monolayer TMDs have a band nesting region, causing van Hove singularities in the joint density of states[30,31].

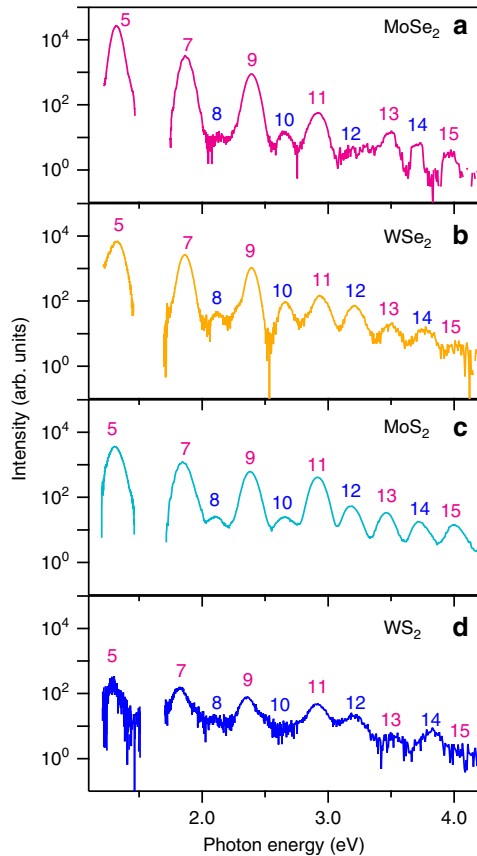

**Fig. 1** High harmonic generation from monolayer TMDs. High harmonic spectra from **a** MoSe$_2$, **b** WSe$_2$, **c** MoS$_2$, and **d** WS$_2$ monolayers at room temperature induced by mid-infrared pulse excitation (photon energy 0.26 eV, peak intensity 1.7 TWcm$^{-2}$). We set the excitation polarization in parallel with the zigzag direction. The harmonics from fifth to sixteenth order are labeled

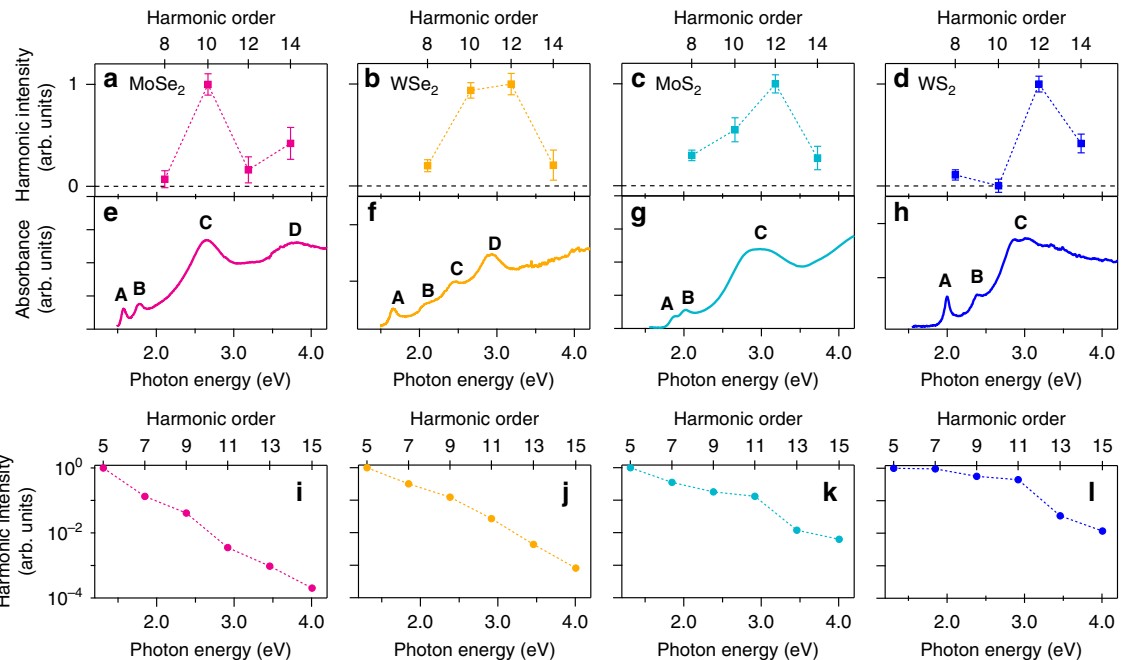

**Fig. 2** Resonant enhancement of even-order high harmonics. Intensities of even-order harmonics in **a** MoSe₂, **b** WSe₂, **c** MoS₂, and **d** WS₂ monolayers. Error bars represent the standard deviations. **e**, **f**, **g**, **h** Corresponding optical absorption spectra of the TMD monolayers. The peaks of the A and B excitons are labeled A and B, respectively. The peaks due to the band nesting effects are labeled C and D. **i**, **j**, **k**, **l** Corresponding intensities of odd-order harmonics

The band nesting results in strong optical absorption at higher photon energies (labeled C and D in the figure)[32–34]. As shown in Fig. 2a, the intensity of the 10th harmonic in monolayer MoSe₂ is larger than that of the 8th, and the intensity of the 14th harmonic is larger than that of the 12th. Interestingly, the energies of the 10th and 14th harmonics coincide with the absorption peaks C and D. This is not only the case in MoSe₂. Figure 2b–d also shows enhancements at the band nesting energies for WSe₂, MoS₂, and WS₂. The observed enhancement in resonance with the interband optical transition strongly suggests that the interband contribution is dominant in even-order HHG. On the other hand, the intensities of the odd orders monotonically decrease with increasing order as shown in Fig. 2i–l. However, though it is not so apparent, the shoulder-like structure appears at the C and D transitions. This could suggest that odd-order harmonics are contributed by both intraband process, which may show no resonant enhancement around interband transitions, and interband process, which should show the resonances. The experimental data imply that, at lower energy region, especially below the bandgap energy, the intraband contribution is much larger than the interband one for the odd-order harmonics, but interband polarization may substantially contributes it at the resonance energies.

**Polarization selection rules of HHG**. The resonant enhancement of HHG indicates that the interband polarization mainly contributes to even-order HHG. The question is how the perpendicular polarization arises in the even-order harmonics. The transverse current due to a material's Berry curvature might not contribute to interband HHG. Here, we will discuss the mechanism of HHG in monolayer TMDs on the basis of a semiclassical three-step model in solids. The discussion is fundamentally the same as the one on the polarization of high-order sideband generation in ref. [34]. In our experiment, the incident light was linearly polarized; i.e., it was a linear combination of left- and right-hand circular polarized light. In the strong-field

regime, therefore, electron–hole pairs are generated coherently by Zener tunneling at the K and K' points in the two-dimensional hexagonal Brillouin zone. We will assume that electron–hole pairs are created when the electric field of the incident light reaches positive or negative maxima. Since the generated electrons and holes are accelerated differently in the conduction and valence bands, their motions are asymmetric in real space (Fig. 3c). When the electron and hole meet again in real space (the circle in Fig. 3d), electron–hole recombination occurs and HHG is emitted. The electron–hole pairs created at the positive peak field ($t =$ 0 in Fig. 3d) under excitation with a polarization along the zigzag direction (Fig. 3a) have anisotropic driving processes: those created at the K point are driven in the K – Γ – K' direction, while those created at the K' point are driven in the K' – M – K direction. This anisotropy in turn causes anisotropic acceleration and recombination dynamics in the K and K' bands ($0<t<T/2$, where T is the period of the incident light). The dynamics of the electron–hole pairs generated at the negative peak field ($t = T/2$) are the reverse of those generated at the positive peak field ($T/2<t<T$). On the other hand, when the excitation light is polarized along the armchair direction, the electron–hole pairs generated at the positive peak field and those generated at the negative peak field have the same dynamics in the K and K' bands (Fig. 3e).

Next, we show how the anisotropic and alternating nature of the electron–hole dynamics gives rise to a polarization selection rule for HHG. The electric field of HHG is the sum of the left- and right-hand circular polarized light:

$$\mathbf{E}^{\mathrm{HHG}}(t) = E_+^{\mathrm{HHG}}(t)\boldsymbol{\sigma}^+ + E_-^{\mathrm{HHG}}(t)\boldsymbol{\sigma}^- \qquad (1)$$

where $E_\pm^{\mathrm{HHG}} = \left(E_x^{\mathrm{HHG}} \pm iE_y^{\mathrm{HHG}}\right)/\sqrt{2}$ are the amplitudes of the $\boldsymbol{\sigma}^\pm$ components. Here, we regard the x axis as the zigzag direction and suppose mid-infrared excitation with the polarization of the zigzag direction: $\mathbf{E}_{\mathrm{zigzag}}^{\mathrm{MID}}(t) = E_x(t)(\boldsymbol{\sigma}^+ + \boldsymbol{\sigma}^-)/\sqrt{2}$. We only consider the ballistic driving process for HHG since the electrons and

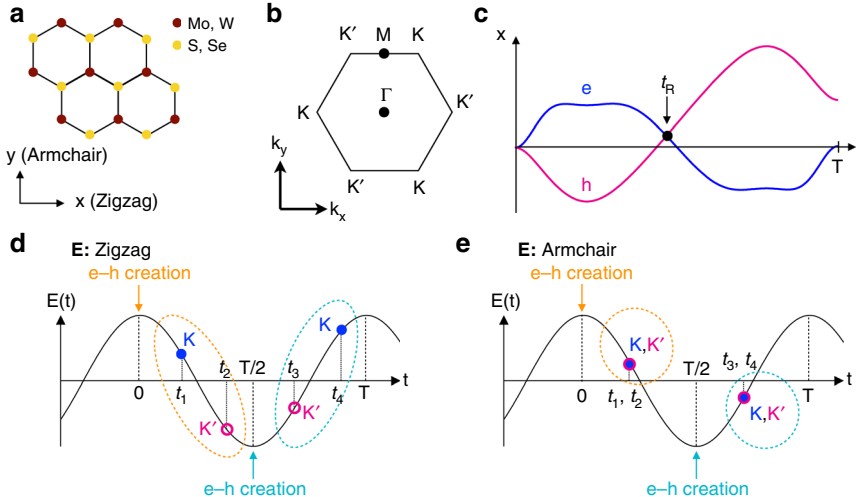

**Fig. 3** electron–hole dynamics of HHG in monolayer TMDs. **a** Top view of the crystal of monolayer TMDs. The brown dots are transition-metal atoms, while the yellow dots are chalcogenide atoms. The $x$ ($y$) axis corresponds to the zigzag (armchair) direction. **b** Two-dimensional hexagonal Brillouin zone of monolayer TMDs. The high symmetry, $\Gamma$, $K$ ($K'$), $M$ points are labeled. The electric field with a polarization in the $x$ ($y$) direction in real space drives carriers in the $k_x$ ($k_y$) direction. **c** Schematic diagrams of the real space trajectories of electrons (blue curve) and holes (magenta curve) driven by light field. The electron–hole pairs generated by Zener tunneling at $t = 0$ are accelerated by the electric field, and they recombine when they meet again ($t = t_R$). **d** Schematic drawings of electron–hole creation and recombination dynamics under excitation with a polarization along the zigzag direction. The electron–hole pairs generated at $t = 0$ follow different traces depending on whether they were created at the $K$ or $K'$ point. Thus, the electron–hole pairs recombine at different times $t_1$ and $t_2$ (blue solid and magenta open circles in the dashed orange oval). The electron–hole pairs generated at $t = T/2$ follow reversed dynamics to those created at $t = 0$ (magenta open and blue solid circles in the dashed turquoise oval). **e** On the other hand, under excitation with the polarization along the armchair direction, the acceleration and recombination dynamics of the electron–hole pairs do not depend on whether the pairs were created at the $K$ point or at the $K'$ point

holes losing their initial coherence do not contribute to high harmonics. We assume that the relative phase between electron–hole pairs generated at $K$ and $K'$ points is conserved in the whole process[19]. Even if their spin states change by driving process, the following discussion is valid when the initially generated, relative phase is not lost. The time-reversal symmetry of the system and the periodicity of the incident light with the polarization of the zigzag direction follows the relation (see Supplementary Note 4 for the detailed discussion):

$$E_{\pm}^{\mathrm{HHG}}(t) = -E_{\mp}^{\mathrm{HHG}}\left(t + \frac{T}{2}\right) \qquad (2)$$

This directly gives the selection rules:

$$E_x^{\mathrm{HHG}}(t) = -E_x^{\mathrm{HHG}}\left(t + \frac{T}{2}\right) \qquad (3)$$

and

$$E_y^{\mathrm{HHG}}(t) = E_y^{\mathrm{HHG}}\left(t + \frac{T}{2}\right) \qquad (4)$$

By considering $\mathbf{E}^{\mathrm{HHG}}(t) = \sum_n \mathbf{E}_n e^{-in\omega t} = \sum_n \left[E_x^n \hat{\mathbf{x}} + E_y^n \hat{\mathbf{y}}\right] e^{-in\omega t}$, equations (3) and (4) tell us that the odd-order HHG has parallel polarization and even-order HHG have perpendicular polarizations to the incident light. The polarization selection rules of the HHG obtained here explain the results of our experiment and ref. [23]. We also discuss the polarization selection rule of HHG with the armchair excitation (see Supplementary Note 4).

**Calculation of dynamics of electrons and holes by semiclassical treatment**. To understand the resonant effect of interband HHG in the band nesting energy region, it is worthwhile to know where the recombined electron–hole pairs are in

momentum space. Here, we calculated the electron–hole dynamics in real and momentum space for monolayer $MoS_2$ on the basis of the three-step model. The valence and conduction band were given by a tight-binding model without spin-orbit couplings[35,36] (see the Supplementary Note 3). We supposed that the incident light in the calculation has a sine-like electric field with a maximum field of $27 MV cm^{-1}$ and a polarization in the zigzag direction (see the Supplementary Note 5 for the case of the polarization in the armchair direction). Moreover, we assumed that the electron–hole pairs are created at the $K$ and $K'$ points when the electric field reaches a peak (at $t = 0$, $T/2$, $T$, … in Fig. 3d) through the Zener tunneling process. The electron–hole pairs are accelerated by the light field and eventually recombine with each other by emitting high-energy photons. When the electron–hole pairs in the $K$ valley are driven in the $K - \Gamma - K'$ direction, those in the $K'$ valley are driven in the $K' - M - K$ direction (Fig. 3b).

Here, we propose an extension of the conventional three-step model that includes quantum mechanical effects for electron–hole pairs in solids: the trajectories of electrons and holes in momentum space are calculated using the equation of motion for Bloch electrons: $\hbar\dot{\mathbf{k}} = q\mathbf{E}$, where $\mathbf{k}$ is the electron's wave vector, $q$ its charge, and $\mathbf{E}$ the electric field. In real space, we should consider a wave packet of Bloch electrons and holes. The equation of motion should be $\dot{\mathbf{r}} = \hbar^{-1}\partial\varepsilon(\mathbf{k})/\partial\mathbf{k}$, where $\varepsilon(\mathbf{k})$ is the electron's energy. It is obvious that the wave packet can occupy any lattice site in real space due to the periodicity. Therefore, we considered several trajectories starting from different atomic sites. Here, we limited our calculation to one dimension because the transverse motion due to the Berry curvature[23] is relatively small and the obtained results do not change qualitatively. Figure 4a, b shows the real-space trajectories of the carriers generated at $t = 0$ at the $K$ ($K'$) point. The turquoise curves indicate the motion of a

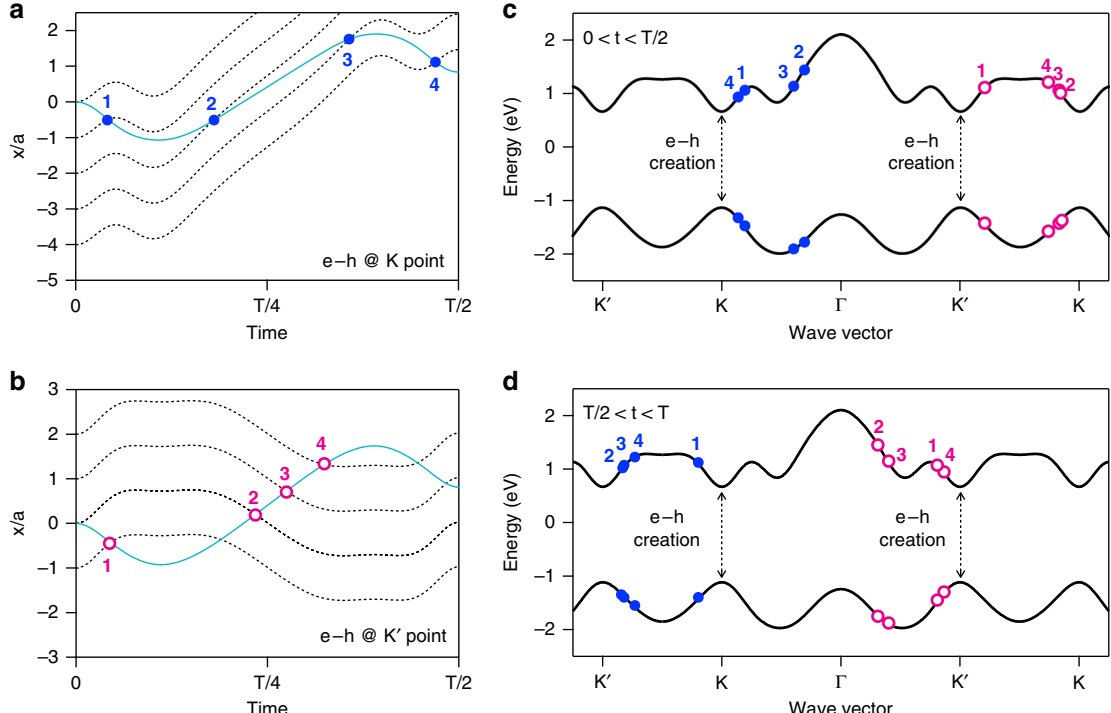

**Fig. 4** Semi-classical calculation of electron–hole dynamics in the real and momentum space. **a** Real space dynamics of carriers generated at $t = 0$ and the $K$ point under excitation with polarization along zigzag direction. The turquoise curve shows the trajectory of the hole generated at $x/a = 0$, and the black dashed curves show the electrons generated at $x/a = 0$, $-1$, $-2$, $-3$, $-4$. The possible recombination paths of the hole and electrons are marked as blue solid circles. **b** Same as **a** but for the carriers generated at the $K'$ point. The possible recombination paths of the hole and electrons are marked as magenta open circles. **c** Recombined electron–hole pairs in momentum space generated at $t = 0$. The electron–hole pairs created at the $K$ ($K'$) point are represented as blue solid (magenta open) circles. The labels from 1 to 4 correspond to those in **a** and **b**. The asymmetric features of the electron–hole dynamics in the $K$ and $K'$ valleys lead to different times and energies of recombination. Only the electron–hole pairs generated at the $K$ point reach the band nesting region. **d** Same as **c** but for electron–hole pairs generated at $t = T/2$. The dynamics in the $K$ and $K'$ valleys are reversed from those of $t = 0$ and show the alternating nature. Only electron–hole pairs generated at the $K$ point reach the band nesting region

hole generated at $x/a = 0$. The black dashed curves show the motion of electrons generated at several neighboring atomic sites around the hole generation site ($x/a = 0$). HHG emissions can happen when an electron and hole meet again in real space. For a hole generated at the $K$ point, there are four ways for it to recombine with the electrons, marked as circles in Fig. 4a. In this case, it should be noted that the hole recombines with an electron generated at a different atomic site. This would be a characteristic feature of HHG in solids. For the hole generated at the $K'$ point, there are also four recombination possibilities, marked as circles in Fig. 4b. By comparing the dynamics in real space with those in momentum space, one can determine the momentum and energy of electron–hole pairs at the recombination time. The circles in Fig. 4c represent the recombined electron–hole pairs generated at $t = 0$ in momentum space. The blue solid (magenta open) circles indicate the electron–hole pairs generated at the $K$ ($K'$) point, which result in an $\sigma^+(\sigma^-)$ emission. The two pairs generated at the $K$ points labeled 2 and 3 are in the band nesting region at the time of recombination. The band nesting causes a large joint density of states in the C and D energy region, as shown in the absorption spectra (Fig. 2a–d). The large joint density of states in turn causes a resonant enhancement of HHG at those energies (Fig. 2e–h). Actual electrons and holes are wave packets with a finite uncertainty in momentum, originating from the uncertainty of the time at which the pair was created via the Zener tunneling process. Thus, if we think about the electron–hole pairs in the band nesting region (magenta open circles labeled 2 and 3) and the others, the former cause light emissions with a narrower

energy distribution than those of the latter, which cause the resonant enhancement in the band nesting energy. Figure 4d shows the same kind of plot as Fig. 4c for the electron–hole pairs generated at $t = T/2$. A half period later, the dynamics in momentum space reverse and the electron–hole pairs generated at the $K'$ point reach the band nesting region. From the viewpoint of the polarization selection rule, the alternating nature of the dynamics in the $K$ and $K'$ valleys at $t = 0$ and $t = T/2$ is important. The interband emission process changes every half period of the incident light; that is, light with the same photon energy but opposite helicity is emitted in the subsequent half period. As discussed in the previous section, the asymmetric nature of the dynamics in the $K$ and $K'$ valleys directly lead to polarization selection rules.

## Discussion

In summary, we investigated generation of high harmonics in monolayer TMDs. By comparing four monolayer TMDs, we found resonant enhancement of HHG with the interband optical transition due to band nesting effects. We also found that odd and even orders of HHG have parallel and perpendicular polarizations under excitation with polarization along with the zigzag direction. This comes from the asymmetric nature of the dynamics in the $K$ and $K'$ valleys. Our findings give the important indication that the nonlinear interband polarization significantly contributes to the high harmonic generation in solids and opens the way for attosecond science with monolayer materials having widely tunable electronic structures.

## Methods

**Samples**. The monolayer TMDs were grown on sapphire substrates by chemical vapor deposition. The monolayer flake size is typically tens of micrometers. The $WS_2$ and $WSe_2$ monolayers were prepared using the method reported in ref. [37], and $MoS_2$ and $MoSe_2$ were purchased from 2d Semiconductors, Inc. The monolayer nature of the TMD samples was confirmed by photoluminescence and Raman spectroscopy.

**Experiments**. Supplementary Figure 1 shows the experimental configuration. A mid-infrared femtosecond laser pulse was generated using a differential frequency generation (DFG) configuration consisting of a combination of a Ti:sapphire based regenerative amplifier (800nm center wavelength, 35fs pulse duration, 1kHz repetition rate, and 1mJ pulse energy) and a β-BaB$_2$O$_4$ (BBO) based optical parametric amplifier (OPA). The photon energies of the signal and idler generated by the OPA can be tuned from 0.84 to 0.98 and from 0.71 to 0.57eV, respectively. The difference frequency of the signal and idler beams was produced in a AgGaS$_2$ crystal, which yielded mid-infrared pulses centered at 0.12–0.41eV. A long-pass filter transparent on the longer wavelength side above 4μm (0.31eV) was used to block the signal and idler beams. For the measurement of the power dependence, the power of the mid-infrared excitation pulses was controlled by using a liquid crystal variable retarder (LCC1113-MIR, Thorlabs) and a ZnSe wire grid polarizer (WP25H-Z, Thorlabs). The available photon energy of the mid-infrared pulses was restricted by the transmission photon energy of the long-pass filter and the variable retarder; it was 0.23–0.29 eV. The mid-infrared excitation pulse was focused on the sample by a ZnSe lens with a 62.5 mm focal length to a spot about 30μm in radius. The spectral linewidth of the excitation pulse was ~60meV in full-width half-maximum (Supplementary Figure 2a). Assuming a Fourier-transform-limited pulse, the pulse duration was estimated to be 25fs. The maximum intensity of the pump pulse at the sample was 1.7TWcm$^{-2}$, corresponding to an electric field of 27MVcm$^{-2}$ inside the monolayer TMDs, when taking into account the reflectivity of the substrate. The generated HHG was collected by a fused-silica lens (f = 50mm) and analyzed by a grating spectrometer (iHR320, Horiba) equipped with a Peltier-cooled charge-coupled device camera (Syncerity CCD, Horiba). The high-harmonic spectra between 1.1 and 4.2eV including the fifth to fifteenth harmonics were corrected for the grating efficiency and the quantum efficiency of the detector. The data around 1.55eV in the spectra are not shown because there is a stray light in the experimental setup caused by the output of Ti:sapphire regenerative amplifier. The full spectrum from the monolayer $MoS_2$ up to 5eV without an efficiency correction over 4.2eV showed harmonics up to eighteenth order (Supplementary Figure 2b). All experiments were performed at room temperature. The optical absorption spectra were measured with a scanning spectrophotometer (UV-3100PC, Shimadzu).

## Data availability

The data that support the findings of this study are available from the corresponding authors upon reasonable request.

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

## Acknowledgements

This work was supported by a Grant-in-Aid for Scientific Research (A) (Grant No. 26247052) and Scienctific Research (S) (Grant No. 17H06124). N.Y. was supported by a JSPS fellowship (Grant No. 16J10537). Y.M. acknowledges support from JST CREST (Grant No. JPMJCR16F3) and a Grant-in-Aid for Scientific Research(B) (Grant No. JP18H01832) from the Ministry of Education, Culture, Sports, Science and Technology (MEXT), Japan.

## Author contributions

N.Y. and K.T. conceived the experiments. N.Y. and K.U. carried out the experiments and analyses. Y.T., S.S. and Y.M. fabricated the samples. K.N. and K.T. performed the calculation. N.Y. and K.T. wrote the manuscript. All the authors contributed to the discussion and interpretation of the results.

## Additional information

**Competing interests:** The authors declare no competing interests.

**Peer review information:** *Nature Communications* would like to thank the anonymous reviewers for their contribution to the peer review of this report. Peer review reports are available.

