## [Peer Review File · Nature Communications]

Reviewers' comments:

Reviewer #1 (Remarks to the Author):

The work of Yoshikawa et al., "Interband resonant high-harmonic generation by valley polarized electron-hole pairs", discusses the process of high harmonic generation in four different semiconducting TMDs (MoS₂, MoSe₂, WS₂, WSe₂) and the origin of the perpendicular polarization with respect to the incident laser for the even order harmonics. The authors observe high harmonics up to the 18th order (although in fig. 1 of the manuscript they show only up to the 15th order, why?) with the intensity of the even order harmonics that is not monotonically decreasing as the harmonic order increases. The observation of high-harmonic generation in TMDs has already been reported. Thus, in my understanding, the main goal of this work is to discriminate whether even order harmonics in TMDs arise from intra- (as discussed in Nat. Phys. 13, 262, 2016) or inter-band processes. The only new experimental results, to support the thesis that even harmonics arise from inter-band transitions, is a comparison between the absorption spectra of the different TMDs and the intensity of even order high-harmonics. This result (fig.2) nicely shows an increase of the harmonics intensity when these are resonant with the electronic transitions in the band-nesting region of the TMDs (C-D transitions). Based on this result, the authors develop a model to explain the origin of the parallel polarization and intensity of even order harmonics without recurring to intra-band transitions.

Overall, the results are interesting and the topic will certainly catch the attention of the readers of Nat. Communications. However, I find the manuscript a bit difficult to read and, in some points, not fully convincing. My suggestion is to resubmit the manuscript after some major revisions:

1. In the first part of the manuscript there is a discussion that I find very difficult to read. One point regards the absence of "exciton luminescence" while HHG exceeding the bandgap was generated. This observation seems to be, according to the authors, useful to discriminate between the inter- and intra-band origin of the HHG. I find the sentence not fully convincing. The idea that inter-band HHG would not leave excited electrons in the conduction band would only work for conversion efficiencies close to 100%, which is clearly not the case here.
2. In fig.2 the authors only show the enhancement of the even harmonics while from fig.1 it seems that the odd harmonics are monotonically decreasing for increasing harmonic order. It would be interesting to compare also the odd-harmonics with the TMDs absorption. Is there any evidence of resonant enhancement at the C and D transitions for the odd order harmonics? The authors should carefully discuss this point.
3. It is not clear if the HHG measurements in fig.2 report the intensity of HHG only for one polarization (perpendicular to the excitation pulse) or for any polarization. By looking at fig. S3 it is clear that the 9th harmonic has only components that are parallel to the excitation pulse, while the 12th harmonic has both parallel and perpendicular components. The presence of a parallel component in the even order harmonics of MoS₂ was already (briefly) discussed in Nat. Phys. 13, 262, 2016, in particular: "Within this picture, however, the presence of a nonzero component polarized parallel to the pump demonstrates that interband transitions must also contribute to the generation process. The need to include the interband response beyond tunnelling is not surprising, given that the optical properties of the monolayer are strongly influenced by excitonic effects." Thus, to make their claims more convincing, the authors should plot fig.2 separately for the two components of the polarization, parallel and perpendicular to the excitation pulse and discuss the results.
4. I think it would be useful to show a microscope image of the samples and clarify how the zigzag and armchair directions were identified.
5. The authors should consider including recent relevant papers for harmonic generation in layered materials. In particular, harmonic generation in MoS₂ has been intensively discussed in [Nat. Comms.

8, 893 (2017)] while the enhancement of THG due to multi-photon inter-band transitions in graphene has been discussed in [Nat. Nanotechnol. 13, 583 (2018)] and [Nat. Photon. 12, 430 (2018)].

Reviewer #2 (Remarks to the Author):

In their manuscript, "Interband resonant high-harmonic generation by valley polarized electron-hole pairs", Yoshikawa et al. present results from high-harmonic generation (HHG) of a sub-resonant MIR laser from a variety of single layer transition metal di-chalcogenides (TMDC). The results show even and odd harmonics up to ~ 15 th order that scale non-perturbatively with the drive intensity when the light is polarized along the Γ -K-K' direction. The results are interpreted in terms of: 1. symmetry arguments, 2. resonant enhancement due to band-nesting and 3. an extended semi-classical picture of high harmonics tunneling, acceleration and recombination that includes the possibility of re-collisions of an electron and a corresponding hole that may differ by an integer lattice constant. The results suggest that the nonlinear interband polarization contributes significantly to HHG in these materials.

The experimental results are very nice and substantially extend the original work Liu et al., (ref 17 on MoS₂) to include multiple TMDCs. In particular, the results of ref 17 are yet to be understood fully, including the mechanism for enhancement of even harmonics at high orders and the role of inter-band contributions to the HHG process.

This in itself wouldn't necessarily be reason for publication in Nat. Comm.; however, the additional insight obtained should be of interest to the broad physics community, as 1. solid-state HHG is a rapidly growing field and 2. the mechanisms involved and how they depend on the microscopic materials properties is still not well understood. To this end, the authors purport an alternate explanation for the polarization of the even harmonics from ref. 17 in terms of inter-band instead of intra-band dynamics, come up with a possible explanation for the details of the even harmonic spectra with respect to electronic band-nesting, and finally they extension of previous semi-classical models HHG to include recombination of displaced electrons and holes could be valuable to the field. Thus in principle, I support publication, but I do have a few concerns which I list below.

1. The possibility of resonant enhancement of harmonics with energies corresponding to van-hove singularities is indeed attractive. However, the results appear more suggestive than an actual demonstration. Thus, sentences such as that starting on p 6 line 8, "the resonant enhancement of HHG indicates that the interband polarization mainly contributes to HHG" seem much too strong and either needs additional justification, or to be qualified (e.g. ...suggests a substantial contribution to HHG). Similar statements are made in the abstract, on p4 lines 14-17, ...

Do the authors see/expect a similar enhancement for the odd harmonics, and if not why? Also, apart from the material differences, it would be much stronger if they could use polarization, intensity or other parameters to control the trajectories away from recombination near the nesting points and show that the possible resonance affects disappear.

I also note that even here the connection is only approximate by comparing the harmonic spectra, which in their model is largely a 1D problem, with the absorbance which included states in the entire 3D Brillouin zone. it would be useful if they could also point to the the regions of folding in Figure 4 (c/d) to show that there is still a large joint-density of states even along the 1D trajectories in reciprocal space.

2. The symmetry analysis is nice but somewhat lengthy and at least for me was distracting. It was not clear however, whether this is something new, or just to help the reader. If it is not essential, I would suggest either moving it out of the main text, or reducing it.

3. The semiclassical model is helpful in that it demonstrates the possibility for recombination for multiple trajectories within reach $1/2$ cycle of the laser. However, it is still only qualitative, as the authors do not calculate a spectrum in their model. The addition of recombination from electrons/holes that were produced from different sites, is potentially very interesting and important. However, for this to happen, there still needs to be both coherence and some overlap. This would depend critically on the spatial extent of the (non-plane-wave part of the) wave functions and of course the transition dipole.

By the way, the authors seem to miss an important reference to the generalized 3-step model initially put forward in Vampa, et al., PRL 113, 073901 (2014).

Some additional points.

4. in the Abstract, the sentence "We extended the three-step model for electron-hole pairs in solids into one that takes account the possibility of a wave packet of Bloch electrons occupying any lattice site in real space" is awkward and possibly misleading. Don't they instead simply assume that the recombination process occurs when the electron and hole find each other in real space, up to the possible addition of a real lattice vector? Also, did they really extend the model, or simply show that such trajectories exist.

5. On p3, lines 7-9 and 9-12 regarding high harmonic spectroscopy/tomographic imaging in gas and probing electronic structure in solids, references need to be added.

6. On p5 lines 9-12, the argument for inter-band contributions based on the lack of exception luminescence being present is much overstated. First, the fraction of electron-hole pairs that recombine may be very small. Second, is it clear that intraband contributions only occur if the electron and hole remain excited after the pulse?

7. P7 lines 15-16, is it clear that the spin states cannot change during rapid acceleration in k-space?

8. p10 line 4. "it is obvious that the wave packet can occupy any lattice site in real space due to quantum effects". why does this have to do with quantum effects? Isn't this just a consequence of the periodicity?

9. What was the size of the flakes.

10. Why is the 6th harmonic missing from all of the data?

Response to recommendations and criticisms:

Reply to the Reviewer 1's comments

- 1) In the first part of the manuscript there is a discussion that I find very difficult to read. One point regards the absence of “exciton luminescence” while HHG exceeding the bandgap was generated. This observation seems to be, according to the authors, useful to discriminate between the inter- and intra-band origin of the HHG. I find the sentence not fully convincing. The idea that inter-band HHG would not leave excited electrons in the conduction band would only work for conversion efficiencies close to 100%, which is clearly not the case here.

As pointed by both Reviewer 1 and 2, the discussion about the relation between the luminescence and HHG in the previous manuscript was overstated. To discuss the relative intensity of luminescence and HHG, the quantitative calculation with a sophisticated theoretical model is necessary. So, we removed this part. It does not change any argument and conclusion in this paper.

Action taken:

We removed the sentences from page 5 line 10.

- 2) In fig.2 the authors only show the enhancement of the even harmonics while from fig.1 it seems that the odd harmonics are monotonically decreasing for increasing harmonic order. It would be interesting to compare also the odd-harmonics with the TMDs absorption. Is there any evidence of resonant enhancement at the C and D transitions for the odd order harmonics? The authors should carefully discuss this point.

This is an important suggestion, and also pointed out by Reviewer 2. In the previous manuscript, we only explained the monotonical decrease of odd-order harmonics from MoSe₂ by one sentence. It was really insufficient for describing behaviors of harmonic intensities. Accordingly, we added the intensity of the odd-order harmonics as Figs. 2(i)-2(l) in the revised manuscript. As pointed out by Reviewer 1, the intensity of odd order harmonics seems to be monotonically decreasing for increasing harmonic order. However, if looking carefully to Figs. 2(i)-2(l), the shoulder-like structures are clear

around the C and D transitions. This suggests that the enhancement should exist even in the odd-order harmonics but is hidden by the contribution from the intraband process. The intraband process may show no resonant enhancement around the band nesting region. The experimental data imply that the intraband contribution is much larger than the interband one for the odd-order harmonics in the low energy region, but the interband polarization substantially contributes it at the resonance energies.

Action taken:

We updated Fig. 2. We modified and added the sentences on page 6 line 5-12.

- 3) It is not clear if the HHG measurements in fig.2 report the intensity of HHG only for one polarization (perpendicular to the excitation pulse) or for any polarization. By looking at fig. S3 it is clear that the 9th harmonic has only components that are parallel to the excitation pulse, while the 12th harmonic has both parallel and perpendicular components. The presence of a parallel component in the even order harmonics of MoS₂ was already (briefly) discussed in Nat. Phys. 13, 262, 2016, in particular: “Within this picture, however, the presence of a nonzero component polarized parallel to the pump demonstrates that interband transitions must also contribute to the generation process. The need to include the interband response beyond tunnelling is not surprising, given that the optical properties of the monolayer are strongly influenced by excitonic effects.” Thus, to make their claims more convincing, the authors should plot fig.2 separately for the two components of the polarization, parallel and perpendicular to the excitation pulse and discuss the results.

We agreed to this comment. In the previous manuscript, we show the polarization-unresolved spectra in Fig. 2 without saying details of polarization. We added the explanation to avoid misleading. As pointed out by Reviewer 1, the parallel component in the even order harmonics of MoS₂ is attributed to the interband transitions in Nat. Phys. 13, 262 (2016). This statement is based on their claim that the perpendicularly polarized even-order harmonics originates from the transverse intraband current due to the material’s Berry curvature. However, our findings of the interband resonance reveal that **the perpendicularly polarized even-order harmonics is derived from the interband nonlinear polarization under the zigzag polarization excitation whereas the parallel polarized even-order harmonics is derived from that under the armchair polarization excitation.** Therefore, our argument is completely different from that in Nat. Phys. 13, 262 (2016).

To carefully discuss the polarization of the even-order harmonics, we experimentally determined the polarization of the emitted high harmonics with both zigzag and armchair excitation and added in the Supplementary Note 1. The updated Supplementary Figure S3 clearly show that the even-order harmonics with the zigzag excitation are perpendicularly polarized to the excitation light, while those with the armchair excitation are parallel. We suspect that the excitation polarization was not exactly zigzag direction and the parallel high harmonics slightly contributed to the polarization state in the previous Supplementary Figure S3. We also added a **microscopic theoretical model for the polarization selection rule of HHG** in the Supplementary Note 4. The simple symmetry analysis discussed in the main text and Supplementary Note 4 shows the parallel component of even-order harmonics is obtained in the armchair polarization excitation. This result indicates that our model based on the three-step model works well.

Action taken:

We updated Supplementary Note 4 from page 10 line 5 to page 15 line 9, and Supplementary Fig. S3 in the Supplementary Information. We modified the sentences in page 5 line 6-12. We modified the sentences on page 5 line 7-13 in Supplementary Information.

- 4) I think it would be useful to show a microscope image of the samples and clarify how the zigzag and armchair directions were identified.

As described in Supplementary Note 1, the CVD samples in this study had many crystal domains of monolayers, and their crystal orientations are not identical. We show in Fig. 1 the harmonic spectra measured at the position on the sample where strong even-order high harmonics were observed; that is, the crystal orientation with respect to the polarization of the incident laser was optimized to maximize the even-order harmonics. We used home-built optical microscope system to look for the monolayer islands roughly and we finally set the measuring sample position by seeing the harmonic spectrum itself. To avoid misleading, we modified the explanation about the experimental methods.

Action taken:

We modified the sentence in page 5 line 6-8 in the main text and the sentence on page 5 line 4-5 in the Supplementary Information.

- 5) The authors should consider including recent relevant papers for harmonic generation

in layered materials. In particular, harmonic generation in MoS₂ has been intensively discussed in [Nat. Comms. 8, 893 (2017)] while the enhancement of THG due to multi-photon inter-band transitions in graphene has been discussed in [Nat. Nanotechnol. 13, 583 (2018)] and [Nat. Photon. 12, 430 (2018)].

We carefully checked the papers which are proposed to add to References by Reviewer 1. They are very nice paper about harmonic generation from monolayer materials, but the studies are limited to the perturbative regime. We think that including these papers is not suitable by considering the context of the present paper, that is, the investigation of non-perturbative high harmonic generation.

Reviewer 1 also gave us the following question in the first part of the comment:

...(although in fig. 1 of the manuscript they show only up to the 15th order, why?)...

We show the spectrum up to 15th harmonics in Fig. 2 because the spectra were corrected only between 1.1 and 4.2 eV for the efficiency of the grating and the detector, as explained in **Methods**. We also show the spectrum up to 18th harmonics in Fig. S2. We added the note in the main text to avoid misleading.

Action taken:

We modified the sentences on page 5 line 4-6.

Reply to the Reviewer 2's comments

1-1) The possibility of resonant enhancement of harmonics with energies corresponding to van-hove singularities is indeed attractive. However, the results appear more suggestive than an actual demonstration. Thus, sentences such as that starting on p 6 line 8, "the resonant enhancement of HHG indicates that the interband polarization mainly contributes to HHG" seem much too strong and either needs additional justification, or to be qualified (e.g. ...suggests a substantial contribution to HHG). Similar statements are made in the abstract, on p4 lines 14-17, ...

The resonant enhancement of even-order harmonics is an evidence that the interband contribution is dominant. However, it is not for all harmonics including odd-order harmonics. As we present in the revised manuscript according to the reviewer's comment, the odd-order harmonics is not necessarily mainly dominated by the interband

contribution while it is affected by the resonance condition. To avoid too strong statement, we modified three sentences which were pointed out by Reviewer 2.

Action taken:

We modified the sentences on page 2 line 5-7, on page 4 line 13-16, and page 6 line 15-17.

1-2) Do the authors see/expect a similar enhancement for the odd harmonics, and if not why? Also, apart from the material differences, it would be much stronger if they could use polarization, intensity or other parameters to control the trajectories away from recombination near the nesting points and show that the possible resonance effects disappear.

This is an important suggestion, and also pointed out by Reviewer 1. In the previous manuscript, we only explained the monotonical decrease of odd-order harmonics from MoSe₂ by one sentence. It was really insufficient for describing behaviors of harmonics intensities. Accordingly, we added the intensity of the odd-order harmonics as Figs. 2(i)-2(l) in the revised manuscript. As pointed out by Reviewer 1, the intensity of odd order harmonics seems to be monotonically decreasing as increasing harmonic order. However, if looking carefully to Figs. 2(i)-2(l), the shoulder-like structures are clear around the C and D transitions. This suggests that the enhancement should exist even in the odd-order harmonics but is hidden by the contribution from the intraband process. The intraband process may show no resonant enhancement around the band nesting region. The experimental data imply that the intraband contribution is much larger than the interband one for the odd-order harmonics in the low energy region, but the interband polarization substantially contributes it at the resonance energies.

As also pointed out by Reviewer 2, it is quite interesting if we can control the HHG by controlling the trajectory by using polarization, but it is beyond the present study and should be a future prospect.

Action taken:

We updated Fig. 2. We modified and added the sentences on page 6 line 6-12.

1-3) I also note that even here the connection is only approximate by comparing the harmonic spectra, which in their model is largely a 1D problem, with the absorbance which included states in the entire 3D Brillouin zone. It would be useful if they could

also point to the the regions of folding in Figure 4 (c/d) to show that there is still a large joint-density of states even along the 1D trajectories in reciprocal space.

According to this comment, we calculated the joint-density of states along the $K - \Gamma - K'$ direction in the reciprocal space. It shows divergence features at around 1.8, 2.8, 2.9, and 3.3 eV. The peaks at 2.8 and 2.9 eV are derived from the band nesting, which corresponds to the C peak in Fig. 2(g). We added Supplementary Fig. S6 in Supplementary Note 3.

Action taken:

We added Supplementary Fig. S6. We also added the sentences on page 9 line 4-9 in Supplementary Information.

- 2) The symmetry analysis is nice but somewhat lengthy and at least for me was distracting. It was not clear however, whether this is something new, or just to help the reader. If it is not essential, I would suggest either moving it out of the main text, or reducing it.

According to the Reviewer 2's suggestion, we move the part of symmetry analysis to Supplemental Information as Supplementary Note 4. Only the essential part is remained in the main text. Supplementary Note 4 includes details of derivation polarization selection rules using microscopic model that are improved a lot compared to the previous manuscript.

Action taken:

We move the part of discussion on page 8 to Supplementary Note 4. Accordingly, we modified the sentence on page 8 line 3-4 and added the sentence on page 8 line 15-16 in the main text. We also updated Supplementary Note 4 from page 10 line 5 to page 15 line 9 in the Supplementary Information.

- 3) The semiclassical model is helpful in that it demonstrates the possibility for recombination for multiple trajectories within reach 1/2 cycle of the laser. However, it is still only qualitative, as the authors do not calculate a spectrum in their model. The addition of recombination from electrons/holes that were produced from different sites, is potentially very interesting and important. However, for this to happen, there still needs to be both coherence and some overlap. This would depend

critically on the spatial extent of the (non-plane-wave part of the) wave functions and of course the transition dipole.

By the way, the authors seem to miss an important reference to the generalized 3-step model initially put forward in Vampa, et al., PRL 113, 073901 (2014).

As pointed out by Reviewer 2, our demonstration based on the semiclassical model is qualitative in the previous manuscript. To understand fully the mechanism, we derived polarization selection rules by the microscopic model using the same formula in the paper of Vampa, et al., PRL 113, 073901 (2014) in the Supplementary Note 4. These results improve the theoretical aspect of the paper. Especially, polarization selection rules deduced in the Supplementary Note 4 fully reproduce the experimental results shown in Supplementary Fig. S3. We believe that combining these considerations with the semi-classical model gives an intuitive insight of HHG. We also added the ref. [19] according to this comment.

Action taken:

We updated Supplementary Note 4 from page 10 line 5 to page 15 line 9 in the Supplementary Information. We added Ref. [19].

- 4) in the Abstract, the sentence "We extended the three-step model for electron-hole pairs in solids into one that takes account the possibility of a wave packet of Bloch electrons occupying any lattice site in real space" is awkward and possibly misleading. Don't they instead simply assume that the recombination process occurs when the electron and hole find each other in real space, up to the possible addition of a real lattice vector? Also, did they really extend the model, or simply show that such trajectories exist.

According to this comment by Reviewer 2, we modified the sentences in the abstract to avoid misleading.

Action taken:

We modified the sentences on page 2 line 9-13.

- 5) On p3, lines 7-9 and 9-12 regarding high harmonic spectroscopy/tomographic imaging in gas and probing electronic structure in solids, references need to be added.

Action taken:

We added Ref. [7] and [13-16]. We also modified the sentences on page 3 line 7-10.

- 6) On p5 lines 9-12, the argument for inter-band contributions based on the lack of exception luminescence being present is much overstated. First, the fraction of electron-hole pairs that recombine may be very small. Second, is it clear that intraband contributions only occur if the electron and hole remain excited after the pulse?

As pointed by both Reviewer 1 and 2, the discussion about the relation between the luminescence and HHG in the previous manuscript was overstated. To discuss the relative intensity of luminescence and HHG, the quantitative calculation with a sophisticated theoretical model is necessary. So, we removed this part. It does not change any argument and conclusion in this paper.

Action taken:

We removed the sentences on page 5 line 10.

- 7) P7 lines 15-16, is it clear that the spin states cannot change during rapid acceleration in k-space?

We only consider the ballistic driving process for HHG since the electrons and holes losing their initial coherence do not contribute to high harmonics. We assume that the relative phase between electron-hole pairs generated at K and K' points is conserved in the whole process¹⁸. Even if their spin states change by driving process, the following discussion is valid when the initially generated, relative phase is not lost. We modified the sentences to make this point clearer.

Action taken:

We modified the sentence from page 7 line 22 to page 8 line 6.

- 8) p10 line 4. "it is obvious that the wave packet can occupy any lattice site in real space due to quantum effects". why does this have to do with quantum effects? Isn't this just a consequence of the periodicity?

As pointed by Reviewer 2, it is a consequence of the periodicity. We modified it.

Action taken:

We modified the sentence on page 9 line 18-20.

9) What was the size of the flakes.

Typical size of the monolayer flakes is tens of micrometers. We added this information in **Method**.

Action taken:

We added the sentence on page 11 line 17.

10) Why is the 6th harmonic missing from all of the data?

The data around 1.55 eV in the spectra are not shown because there is a stray light in our experimental setup due to the output of Ti:sapphire regenerative amplifier, which was used to generate the mid-infrared excitation pulses. We added the sentence to explain this point in **Method**.

Action taken:

We added a sentence on page 13 line 1-2.

Reviewers' comments:

Reviewer #1 (Remarks to the Author):

The authors' reply to my comments are convincing and the updated version of the manuscript is satisfactory. I therefore support its publication.

Reviewer #2 (Remarks to the Author):

I appreciate the care that went into the authors responses. In my opinion the they have adequately addressed the comments of both referees (which had a remarkable level of overlap). However, I do still have a one concern that the authors should consider before publication.

In particular, while their argument that the even harmonics are dominated by interband contributions and the odd harmonics are dominated by intraband do seem to be consistent with their results as presented, as far as I can tell they didn't calculate the relative roles. What physically in the band-structure is it that makes the evens dominated by interband and the odds by intraband? Naively, I would think that the interband contribution from the even harmonics would be suppressed due to the finite Berry curvature and resultant anomalous currents. In fact the authors seem to ignore the effect of the Berry curvature altogether. Have the authors considered the effect on the real space trajectories and whether they could actually prevent re-collisions? Maybe I am missing something simple here. It would be good if they authors could comment on whether this is an open question or predicted in their model. In any event, the authors need to justify why a 1D approach is sufficient here.

Apart from this issue, in my opinion the manuscript is much improved and worthy of publication in Nat. Comm.

Response to recommendations and criticisms:

Reply to the Reviewer 2's comments

- 1) In particular, while their argument that the even harmonics are dominated by interband contributions and the odd harmonics are dominated by intraband do seem to be consistent with their results as presented, as far as I can tell they didn't calculate the relative roles. What physically in the band-structure is it that makes the evens dominated by interband and the odds by intraband?

At first, we would like to notice that there is a significant interband contribution existing in the higher odd-order harmonics such as 9th, 11th, and 13th orders, where a signature of the enhancement is confirmed around the C and D resonances (Figs. 2 (i)-(l)). This should come from the same mechanism as the even-order harmonics which is explained by the semi-classical model shown in Fig.4 in the main text. We believe that the odd-order harmonics in the lower energy region, especially below the bandgap energy, are dominated by the intraband contribution. This is reasonable because the interband contributions should be small at below the bandgap energy due to no resonance. To make this point clearer, we modified the explanation in the main text. The main issue of this paper is the importance of the interband contribution to the enhancement around C and D resonances and selection rules. Therefore, it is not necessary to calculate the relative roles of intraband and interband contributions as pointed out by the Reviewer 2 in this paper. The calculation needs more sophisticated theoretical model including more detailed band information such as wave functions and transition dipole moments in order to compare these contributions quantitatively and will be the future problem in this line.

Action taken:

We modified the sentence on page 6 line 7-10.

- 2) Naively, I would think that the interband contribution from the even harmonics would be suppressed due to the finite Berry curvature and resultant anomalous currents. In fact the authors seem to ignore the effect of the Berry curvature altogether. Have the authors considered the effect on the real space trajectories and whether they could actually prevent re-collisions? Maybe I am missing something simple here. It would be good if they authors could comment on whether this is an open question or

predicted in their model. In any event, the authors need to justify why a 1D approach is sufficient here.

We would like to thank the Reviewer to state the effect of the Berry curvature for the real space trajectories. In the zigzag (x) excitation, group velocities of the electron and hole created at K or K' point are directed only to x direction. In contrast, the velocities due to the Berry curvature are directed to the perpendicular direction (y). Therefore, in principle, we should solve the re-collision problem in the 2D real space. **Figure R1** shows an example of the 2D trajectories of the electron and the hole *created at the same lattice point* under the zigzag excitation in our experimental condition (the wavelength 4.8 μm and peak electric field 27 MV/cm). Here the Berry curvatures are taken from the reference [R1]. *This figure clearly shows no re-collisions within the half cycle.* However, it should be noticed that the transverse distance (v/a) of the electron and the hole is less than the distance of the neighboring atomic sites while the distance along x/a reaches the several times longer than one, suggesting possible re-collision solutions for electron-hole pairs created at different sites. We applied the same method to search for the re-collisions in 2D real space as described in **Supplementary Note 5** and **Supplementary Figure S7** for the armchair excitation. As shown in **Fig. R2a**, we obtained 5 re-collisions in the momentum space. This result leads to the same conclusion as discussed in the main text with Fig. 4: Three pairs generated at the K points labeled 2, 3 and 4 are in the band nesting region at the time of recombination: The band nesting causes a large joint density of states in the C and D energy region and should give an enhancement of the harmonics. Therefore, we decided to neglect the effect of the Berry curvature for simplicity in this paper and calculated 1D trajectories using the semiclassical model. To make this point clear, we added the sentence in the discussion part and modified **Supplementary Note 5** and **Supplementary Figure S7**.

Figure R1 2D trajectories of the electron and the hole induced in the half cycle of the mid-infrared light excitation (the wavelength 4.8 μm and peak electric field 27 MV/cm). The polarization of the excitation light is the zigzag direction of the crystal.

Figure R2 a Recombined electron-hole pairs in the momentum space created at the K point at $t=0$. The labels from 1 to 5 correspond to re-collisions between the hole created at the origin and the electrons created at 1 to 5 sites shown in (b). **b** Initial positions of the electrons and the hole (fixed at the origin) that make re-collisions with the limited collision time of a half cycle of the incident light field ($0 < t < T/2$). We regard the re-collision of the electron and hole as the pair separation distance becomes smaller than $0.2a$ (a =lattice constant). In the calculation, we used 27 MV/cm for the peak field of the mid-infrared light (the wavelength 4.8 μm) and the zigzag direction for the polarization.

Action taken:

We added the sentence on page 9 line 18-19 and modified **Supplementary Note 5** and **Supplementary Figure S7**.

Reference for the response

[R1] Ting Cao, Gang Wang, Wenpeng Han, Huiqi Ye, Chuanrui Zhu, Junren Shi, Qian Niu, Pingheng Tan, Enge Wang, Baoli Liu and Ji Feng, Valley-selective circular dichroism of monolayer molybdenum disulphide, Nature Communications 3, 887 (2012).

REVIEWERS' COMMENTS:

Reviewer #2 (Remarks to the Author):

I appreciate the authors considering my "last-minute" concerns about using a 1D model for their calculations and for revising the manuscript and supplemental to address it. While I am not completely convinced, I think that what they now write is both a reasonable and transparent way to address my concerns. I am pleased to support publication of their work in Nature Comms.

Response to recommendations and criticisms:

Reply to the Reviewer 2's comments

- 1) I appreciate the authors considering my "last-minute" concerns about using a 1D model for their calculations and for revising the manuscript and supplemental to address it. While I am not completely convinced, I think that what they now write is both a reasonable and transparent way to address my concerns. I am pleased to support publication of their work in Nature Comms.

We appreciate the reviewer's comment and the support of the publication of our work.